# Comparing UNet configurations for anthropogenic geomorphic feature extraction from land surface parameters

**Sarah Farhadpour, Aaron E. Maxwell** *

West Virginia University Department of Geology and Geography, Morgantown, West Virginia, United States of America

* Aaron.Maxwell@mail.wvu.edu

## Abstract

The application of deep learning for semantic segmentation has revolutionized image analysis, particularly in the geospatial and medical fields. UNet, an encoder-decoder architecture, has been suggested to be particularly effective. However, limitations such as small sample sizes and class imbalance in anthropogenic geomorphic feature extraction tasks have necessitated the exploration of advanced modifications to improve model performance. This study investigates a variety of architectural modifications to base UNet including replacing the rectified linear unit (ReLU) activation function with leaky ReLU or swish; incorporating residual connections within the encoder blocks, decoder blocks, and bottleneck; inserting squeeze and excitation modules into the encoder or attention gate modules along the skip connections; replacing the default bottleneck layer with one that incorporates dilated convolution; and using a MobileNetV2 architecture as an encoder backbone. Unique geomorphic datasets derived from high spatial resolution lidar data were used to evaluate the performance of these modified UNet architectures on the tasks of mapping agricultural terraces, mine benches, and valley fill faces. The results were further analyzed across varying training sample sizes (50, 100, 250, 500, and the full training set). Our results suggest that the incorporation of advanced modules can enhance segmentation performance, particularly in scenarios involving limited training data or complex geomorphic landscapes. However, differences were minimal when larger training set sizes were used (e.g., above 500 image chips) and the base UNet architecture was generally adequate. This research contributes valuable insights into the optimization of UNet-based models for anthropogenic geomorphic feature extraction and provides a foundation for future work aimed at improving the accuracy and efficiency of deep learning approaches in geospatial applications. We argue that one of the positive attributes of UNet is that it can be treated as a general framework that can easily be modified.

**Data availability statement:** The terraceDL (https://doi.org/10.6084/m9.figshare.22320373.v2), vfillDL (https://doi.org/10.6084/m9.figshare.22318522.v2), and mineBenchDL (https://doi.org/10.6084/m9.figshare.26042920.v1) datasets are available via FigShare (https://figshare.com/).

**Funding:** Funding was provided by the National Science Foundation (Federal Award ID No. 2046059: "CAREER: Mapping Anthropocene Geomorphology with Deep Learning, Big Data Spatial Analytics, and LiDAR"). Any opinions, findings, and conclusions or recommendations expressed in this material are those of the author(s) and do not necessarily reflect the views of the National Science Foundation.

**Competing interests:** The authors have declared that no competing interests exist.

## Introduction

The UNet semantic segmentation deep learning (DL) architecture [1] has become one of the most popular CNN-based architectures for pixel-level classification. One positive attribute of this architecture is that it can be easily augmented. For example, the traditional encoder can be replaced with the convolutional neural network (CNN) component of a scene classification DL architecture, such as one from the ResNet [2], DenseNet [3], InceptionNet [4], VGGNet [5], or MobileNet [6,7] family. This can allow for easier implementation of transfer learning since these architectures have already been trained on large datasets, such as ImageNet. Other modifications include using alternative activation functions; incorporating dilated convolution; replacing the bottleneck layer; or inserting additional modules into the encoder or decoder or along the skip connections [8,9]. The base UNet architecture has also more recently been augmented to incorporate transformer- [10–12] and/or mamba-based [13,14] modules. We argue that UNet should be conceptualized as a general and customizable framework for performing semantic segmentation, or pixel-level, classification.

The goal of this study is to explore classification performance between modifications of the base UNet architecture across varying training set sizes. We specifically focus on augmentations that do not require the use of transformers or mamba components, as this is a focus of a separate study. In other words, this study focuses on CNN-based architectural changes or components specifically. This work is also specific to the application of extracting geomorphic features using land surface parameters (LSPs), or digital terrain variables, derived from high spatial resolution light detection and ranging (lidar)-derived digital terrain models (DTMs). Such techniques are useful for mapping anthropogenic and archeological features [15–17], conducting surficial geologic mapping [18–20] and landform delineation, digital soil mapping [21–23], and geohazard risk assessment and monitoring [24–26].

In this study, we specifically evaluate the following enhancements: replacing the rectified linear unit (ReLU) activation function with leaky ReLU or swish, incorporating residual connections into the double convolution blocks within the encoder and decoder, replacing the bottleneck block with a dilated convolution (DC) module, incorporating squeeze and excitation (SE) modules into the encoder, adding attention gate (AG) modules along the skip connections between the encoder and decoder, and the use of the CNN component of the MobileNetV2 [6] architecture as the encoder backbone. These different configurations are implemented in the recently released geodl [27] R [28] package, which was created by the authors of this study and implements a configurable UNet architecture, including the ability to incorporate the changes or modules explored in this study.

The architectural configurations are applied to three anthropogenic geomorphic feature extraction tasks: mapping agricultural terraces in Iowa, USA with the terraceDL dataset [29]; identifying valley fill faces resulting from surface coal mining reclamation in southern West Virginia, eastern Kentucky, and southwestern Virginia, USA using the vfillDL dataset [30]; and detecting mine benches resulting from historic

surface coal mining in northcentral West Virginia, USA with the minebenchDL dataset [31]. We also compare model performance across various training set sizes: 50, 100, 250, 500, and all available 512 × 512 pixel image chips.

## Background

### UNet architecture

UNet [1], conceptualized in Fig 1, is an encoder-decoder, CNN-based semantic segmentation architecture with a "U" shape [32]. The encoder generates feature maps at various spatial scales by passing input data or prior feature maps through two 3 × 3 2D convolutional layers per block (i.e., double convolution blocks), each of which is followed by a batch normalization layer and a rectified linear unit (ReLU) activation.

It has been suggested that batch normalization addresses internal covariate shift, enabling larger learning rates and faster convergence while also helping the network bypass local minima [32–35]. ReLU introduces necessary non-linearity, converting negative activations to zero [36–38] and maintaining positive activations [39]. The encoder reduces feature map size in the spatial dimensions through 2 × 2 2D max pooling with a stride of two after each encoder block, resulting in reducing the size of the spatial dimensions by half between each block and by a factor of 16 between the input layer and the bottleneck block [1] when four encoder blocks are used before the bottleneck block. For example, an input image with a spatial resolution of 256 × 256 cells would be reduced to a resolution of 16 × 16 before being passed to the bottleneck block (256 × 256 → 128 × 128 → 64 × 64 → 32 × 32 → 16 × 16). The decoder then restores the spatial dimensions to that of the input data using 2 × 2 2D transpose convolution with a stride of two, or alternatively upsampling methods, such as bilinear interpolation [39]. Skip connections in the UNet architecture concatenate output feature maps from each encoder block with those processed by the preceding decoder block, following the application of upsampling, before passing the

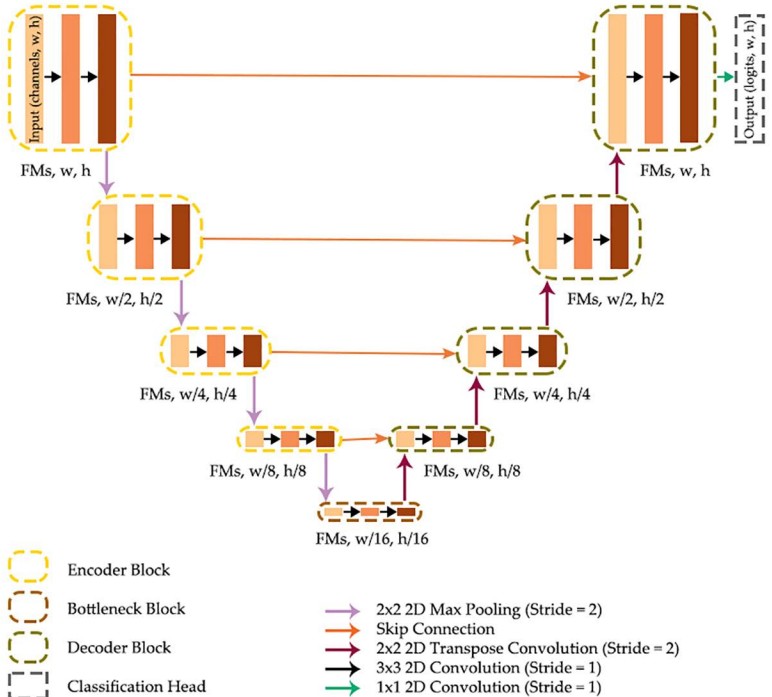

**Fig 1. UNet architecture [ 1].** FMs = feature maps, w = width, h = height. Arrows indicate operations while blocks indicate data: input predictor variables, intermediate feature maps, or output class logits.

feature maps to the decoder block that processes data with the same spatial resolution or at the same level in the architecture. Skip connections enhance information flow and bridge the semantic gap between the encoder and decoder [32].

The feature maps produced by the final decoder block serve as input to the classification head, which uses a 1 × 1 2D convolutional layer to process the feature map values at each pixel location to class logits. For multiclass segmentation, the number of output channels matches the number of classes, while in binary segmentation tasks a single logit for the positive class or a logit for both the positive and background class can be produced.

UNet is widely used in geospatial and remote sensing applications due to its ability to capture fine-scale spatial patterns and maintain high segmentation accuracy across diverse landscapes. For example, Redoloza et al. [40] successfully used UNet to delineate subsurface agricultural drainage from satellite imagery, demonstrating its capability in agricultural landscapes. Dui et al. [41] applied an improved UNet model for automatic detection of photovoltaic facilities from Sentinel-2 MSI imagery, highlighting its effectiveness in multispectral segmentation. In urban studies, Aghayari et al. [42] and Zhao et al. [43] employed UNet-based architectures for building extraction from aerial imagery, emphasizing its robustness to varying object characteristics in remotely sensed data. In ecological applications, Wagner et al. [44] used UNet to map canopy palms in the Amazon rainforest, showcasing its strength in identifying vegetation patterns from high-resolution imagery. Jiao et al. [45] refined UNet for cloud and shadow segmentation in Landsat imagery, addressing limitations in traditional cloud detection techniques. Additionally, Abdollahi et al. [46] leveraged UNet for road segmentation from aerial imagery, further illustrating its adaptability in geospatial feature extraction. Despite these varied applications, relatively few studies have systematically compared multiple UNet variants across tasks within a unified framework. This study aims to fill that gap by evaluating key architectural modifications for geomorphic feature segmentation.

## Activation functions

Activation functions introduce non-linearity into DL architectures and are essential for learning complex mappings between inputs and outputs [38]. Sharma et al. [38] provides a comprehensive review of various activation functions used within neural network architectures and highlights that the ReLU activation is currently the most widely used activation function due to its computational efficiency and effectiveness, despite the issue of dead neurons (i.e., the "dying ReLU problem"). They also note that there is not a consistently best performing activation function, making the choice context-dependent and necessitating experimentation for different tasks. For example, Dubey and Jain [47] investigated the performance of ReLU and leaky ReLU, which maintains negative activations but with a reduced magnitude, within a simple CNN architecture applied to the MNIST (Modified National Institute of Standards and Technology) handwritten digits dataset. ReLU outperformed leaky ReLU in terms of overall accuracy and loss; however, leaky ReLU yielded better recall and F1-score. Thus, defining the optimal activation function for a task can depend on the performance metric of interest.

In the search for optimal activation functions, several new variants have been proposed. Ramachandran et al. [48] employed a combination of exhaustive search and reinforcement learning-based search techniques to discover novel activation functions. The study documented that the swish activation function ($f(x) = x \cdot sigmoid(\beta x)$) consistently outperforms the widely used ReLU function when used within DL models and across various challenging datasets, notably improving top-1 classification accuracy on ImageNet by up to 0.9%. This activation function multiplies the output activation by the activation with a sigmoid function applied, and a $\beta$ term can be multiplied by the activation prior to applying the sigmoid transformation to control its impact on the final transformation.

## Residual connections

Residual connections, introduced as part of the ResNet [2] family of architectures, have been proposed as a means to mitigate the problem of vanishing gradients and enable the training of deeper architectures [2]. As conceptualized in Fig 2, the core idea is to create shortcut connections that bypass one or more model components. These connections add the input of a module to its output, enabling the network to learn residual functions instead of directly learning unreferenced

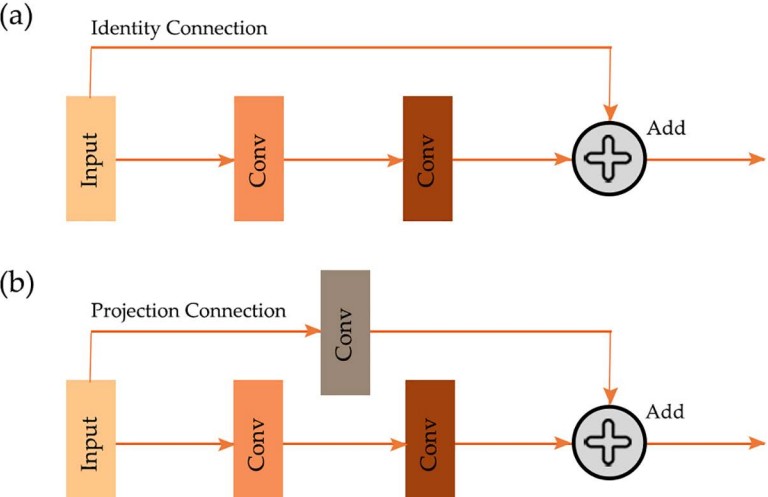

**Fig 2. Residual connection [ 2] within a double-convolution block using an identity connection (a) and projection connection (b).**

mappings. When the number of feature maps and the spatial resolution of the input data are maintained within the block, the input can simply be added to the output. This is known as an identity connection. However, if the number of feature maps and/or spatial resolution change within the block, the input data must be modified along the skip connection. This type of connection is generally termed a projection connection. Along the projection connection, convolution with a stride larger than one can be used to change the spatial resolution of the data and/or number of feature maps while convolution operations with a stride of one can be used to change the number of feature maps but maintain the spatial dimension sizes [2].

Cao and Zhang [49] integrated ResNet into the UNet semantic segmentation encoder architecture in an attempt to improve the classification of tree species using airborne high spatial resolution images. This hybrid architecture, termed Res-UNet, demonstrated superior performance in comparison to traditional UNet. Similarly, Diakogiannis et al. [50] proposed ResUNet-a, which combines a UNet encoder/decoder backbone with residual connections and additional modifications. This architecture achieved an average class F1-score of 92.9% for urban object classification and segmentation from high spatial resolution imagery. Liu et al. [51] documented improved performance for geologic fault mapping from seismic data when using a ResUNet model in comparison to base UNet. Khan et al. [52] proposed RMS-UNet, an architecture that incorporates residual blocks and dilated convolution, which again outperformed base UNet.

These studies collectively highlight the effectiveness of integrating residual connections into CNN architectures. Incorporating residual connections into the UNet architecture helps address the challenges associated with training deep architectures and enhances the model's ability to capture and propagate features.

## Dilated convolution (DC) module

Incorporating dilated convolution, as conceptualized in Fig 3, is another enhancement technique that has been explored to potentially improve the performance of the UNet architecture. In a UNet model, traditional convolution operations use standard 3x3 kernels to process input data, directly comparing a cell with its immediate neighbors. This approach, while effective, can be limited when attempting to capture broader contextual information. Atrous convolution, also known as dilated convolution, as implemented in architectures such as DeepLabv3+ [53], introduces "holes" or zeros into the convolutional kernels. This technique expands the receptive field without increasing the kernel size and number of associated

trainable parameters [32,54,55]. For example, a dilation rate of three inserts two zeros between each pair of kernel elements, enabling the model to consider a wider spatial context without downsampling the feature maps.

Fig 3 conceptualizes the DC module implemented in geodl, which can be used as a replacement for the default double-convolution bottleneck layer. Input data are provided to five 3 × 3 2D convolution layers with user-defined dilation rates, which are set at 1, 2, 4, 8, and 16 by default in the package. Applying atrous convolution at multiple scales simultaneously allows for efficiently capturing spatial context while obviating the need for computationally expensive image pyramids [32,53,55]. The resulting feature maps are then concatenated to a single tensor. Lastly, and in order to change the number of total feature maps generated by the module and passed to the next layer in the architecture, 1 × 1 2D convolution is applied. This method is particularly beneficial in semantic segmentation tasks, as it maintains spatial resolution while capturing multiscale information [55].

The DC module implemented in geodl represents a modification of the atrous spatial pyramid pooling (ASPP) module used within DeepLabv3+, which was proposed by Chen et al. [53] and is a semantic segmentation encoder-decoder structure that leverages dilated convolution in the encoder backbone and incorporates the ASPP module in an attempt to capture multi-scale contextual information and refine predicted object boundaries. Since its inception, DeepLabv3+ has become a commonly used default architecture for a wide range of tasks; however, dilated convolution has also been integrated into UNet. Qiu [56] proposed an enhanced UNet model, integrating ASPP and attention mechanisms with an EfficientNet-B0 encoder to improve segmentation of COVID-19 lesions in CT images. The ASPP module significantly reduced information loss during down-sampling, leading to enhanced segmentation accuracy, sensitivity, and Dice similarity coefficient compared to traditional approaches. In another study, Wang et al. [57] proposed an enhanced UNet model that integrates the coordinate attention mechanism with the ASPP module to improve segmentation accuracy and edge detail processing. The modified model outperformed the traditional UNet based on mean Intersection over Union (mIoU).

## Squeeze and excitation (SE)

Traditional convolutional layers primarily focus on characterizing local spatial patterns as opposed to modeling interrelationships between input channels or learned feature maps. To address this, Hu et al. [58] proposed SE modules, which are designed to characterize channel interdependencies and complement convolutional operations. As shown in Fig 4, the SE module begins with the "squeeze" step, where each input channel is reduced to a single value via global average pooling, resulting in a vector of channel means. These means are then processed in the "excitation" step, where they are passed through a fully connected layer, ReLU activation, and another fully connected layer to model non-linear interrelationships. The output is rescaled using a sigmoid function, and the original input channels or feature maps are multiplied by these rescaled values, enhancing the feature maps based on their importance.

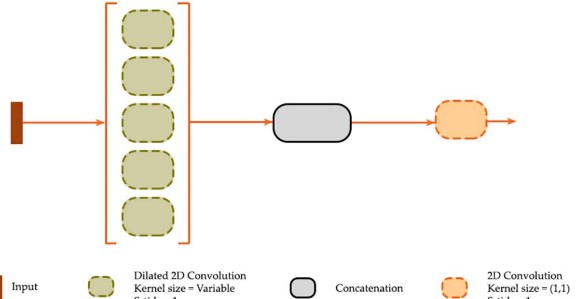

**Fig 3. Dilated convolution (DC) module implemented in geodl [ 27].**

SE blocks have been integrated into UNet; for example, Rundo et al. [59] proposed USE-Net, which incorporates SE blocks into UNet to improve prostate zonal segmentation from MRI datasets, while Üzen et al. [60] proposed a depth-wise squeeze and excitation block-based Efficient-UNet (DSEB-EUNet) model for automatic surface defect detection. In short, the SE mechanism can enrich UNet's feature representations by emphasizing interrelationships between input channels or intermediate feature maps.

## Attention gate (AG)

Attention gate (AG) modules are mechanisms designed to improve the architecture's ability to focus on relevant features in input images while ignoring irrelevant information. An AG enhances feature maps by incorporating semantic information from deeper layers in the architecture. This is illustrated in Fig 5, where feature maps (x) at higher resolution are augmented using a gating signal (g) from lower resolution but deeper feature maps. To align resolutions, x is downsampled using 1 × 1 2D convolution with a stride of two, while g undergoes 1 × 1 convolution with a stride of one to match the number of feature maps in x. The combined result is modified by a ReLU activation, reduced to a single feature map via another 1 × 1 convolution, then rescaled with a sigmoid function. Following upsampling to the original resolution of x using interpolation, the gating signal is multiplied by x to enhance the original signal and highlight key contextual information. The aim of the AG module is to refine feature maps using richer semantic information from later stages in the architecture [61,62].

Several studies have explored different approaches to incorporate AG and similar modules. Oktay et al. [61] proposed a UNet architecture with integrated AG modules for medical image segmentation, which, in comparison to base UNet, improved prediction performance across different datasets and training sample sizes while maintaining computational efficiency. John and Zhang [63] implemented and analyzed an Attention UNet for semantic segmentation of Sentinel-2

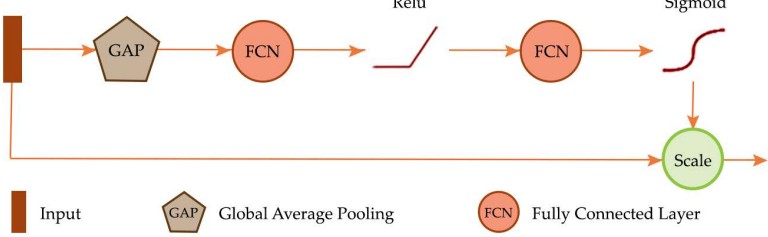

**Fig 4. Squeeze and excitation (SE) module [ 58].**

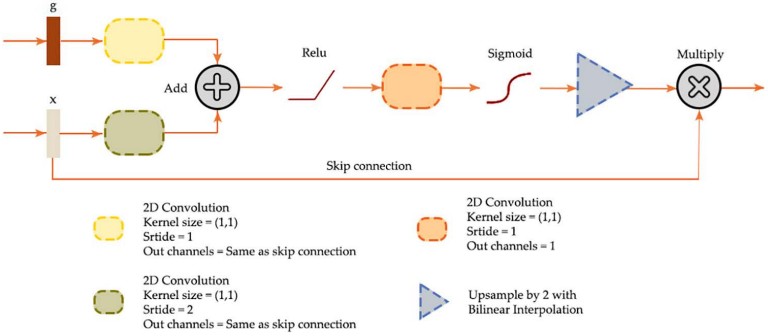

**Fig 5. AG module mechanism [ 61].**

Multispectral Instrument (MSI) satellite imagery to detect deforestation. They compared its performance with UNet and documented improved F1-scores. Yu et al. [64] proposed the AGs-UNet model, which integrated AG modules into a standard UNet architecture to enhance the extraction of building features from high spatial resolution images. The model was validated on two datasets and significantly improved the accuracy and quality of building extraction compared to traditional UNet. Fan et al. [65] developed an improved UNet remote sensing classification algorithm, SA-UNet, which integrated ASPP modules within convolutional units of the encoder and a spatial attention module to enhance the combination of spatial and semantic information.

## Summary of UNet architectures explored

Fig 6 summarizes the UNet architecture explored in this study and implemented within the geodl [27] R [28] package. The base architecture includes four encoder blocks, a bottleneck block, and four decoder blocks. Downsampling in the encoder uses 2×2 max pooling with a stride of two while upsampling in the decoder uses 2×2 transpose convolution with a stride of two. The user can specify the number of output feature maps generated by each block and whether to use residual connections within each block. The ReLU activation function, used throughout the architecture by default, can be replaced with leaky ReLU or swish. When leaky ReLU is used, the negative slope term can be specified. SE modules can be added between the encoder blocks while AG modules can be added along the skip connections. Lastly, the default bottleneck block, which consists of the same double-convolution block used for the encoder and decoder blocks, can be replaced with a DC module, and the user can specify the dilation rates used, the number of feature maps produced by each block, and the final number of feature maps produced after the application of 1×1 convolution.

## MobileNetV2-based encoder

Since the encoder has a similar purpose and structure as the CNN-component of a scene labeling architecture (e.g., varying configurations within the ResNet [2], InceptionNet [4], VGGNet [5], and MobileNet [6,7] families), such architectures

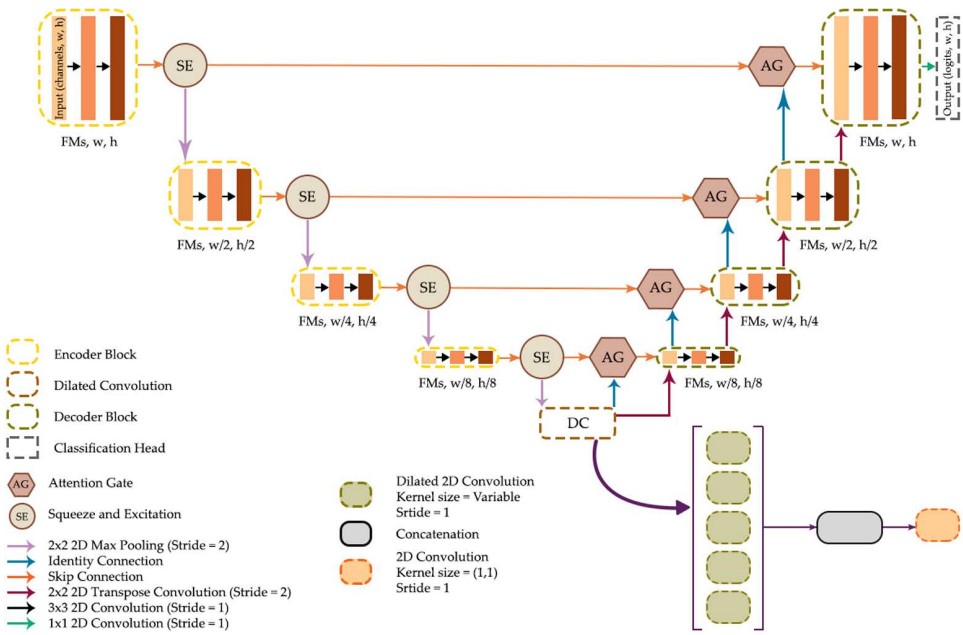

**Fig 6. UNet implementation provided by geodl [27].**

can be used as the encoder component of semantic segmentation architectures [66]. Leveraging powerful CNN back-bones as encoders provides several advantages, including improved feature extraction and the ability to use pre-trained weights, enabling models to benefit from large datasets and enhancing performance when training with limited data [66]. By integrating robust encoders, UNet architectures can achieve higher accuracy and better generalization across various tasks. For example, using EfficientNet- and ResNet-based encoders within UNet architectures has been documented to provide significant improvements in chest X-ray pneumothorax segmentation [67].

In this study, we specifically explore the use of a MobileNetV2-based encoder as opposed to UNet's base decoder. We primarily included this architecture in order to compare the simpler modifications discussed above with a more complex modification: changing the structure of the entire encoder. Further, MobileNetV2 has been incorporated into semantic segmentation architectures in prior studies; for example, both Li et al. [68] and He et al. [69] incorporated it into modified versions of the DeepLabv3+architecture designed for use in remote sensing-related tasks. It was introduced by Sandler et al. [6] and is a lightweight and efficient CNN architecture designed for mobile and edge devices and other resource-constrained environments [6]. Specifically, it incorporates depth-wise separable convolution and inverted residuals with a linear bottleneck to reduce trainable parameters, accelerate convergence rates, and preserve accuracy relative to more complex and computationally expensive architectures [70].

Traditional convolutions apply learned filters to all input channels at once, resulting in a high computational cost. In contrast, depth-wise separable convolutions split this process into two steps: depth-wise convolution and point-wise convolution [71]. The depth-wise convolution applies a single filter to each input channel or feature map separately, and the point-wise convolution then combines these outputs using 1x1 convolution. This separation significantly reduces the number of parameters and computational complexity, making the model faster and lighter without sacrificing accuracy [72]. The adaptability and efficiency of MobileNetV2 have been crucial in achieving high accuracy while ensuring computational efficiency, making it a widely applicable solution for feature extraction in image processing tasks [73]. In this study we use a UNet implementation with a MobileNetV2 encoder provided by the geodl [27] R [28] package.

## Methods

### Study areas and datasets

The data used in this study are described in Fig 7. The terraceDL dataset [29] (Fig 7(a), (d), and (g)) originated from the Iowa Best Management Practices (BMP) Mapping Project [74], where agricultural terrace data were generated as line vector features through manual digitization and interpretation of lidar-derived topographic data and aerial orthophotography. These terraces serve as intentional modifications to the landscape, aimed at mitigating agricultural-induced soil loss by impeding both sheet and rill erosion as well as gully formation. Fig 7(a) illustrates the geographic regions within Iowa, USA used for the study's training, validation, and testing phases, demarcated by hydrologic unit code (HUC) 8 watershed boundaries and containing 66,000, 15,505, and 26,453 terraces, respectively. Fig 7(d) and (g) show example features on the landscape. The terrain representation used in the figures are discussed below.

The vfillDL dataset [30] (Fig 7(b), (e), and (h)) was created by our lab group using manual digitizing and interpretation of digital terrain models and supplemental data. Valley fills are man-made landforms resulting from surface coal mine reclamation. Although the federally implemented Surface Mine Control and Reclamation Act (SMCRA) in the United States mandates restoring mine sites to their "approximate original contour," the steep terrain of southern West Virginia, eastern Kentucky, and southwestern Virginia makes this restoration impractical. Consequently, mining operations are allowed to modify the landscape by leveling mountaintops and depositing overburden material in nearby valleys, creating valley fills [75–78]. As illustrated in Fig 7(b), a large area within West Virginia, which contained 1,105 valley fills, was used to train the model. Neighboring regions in West Virginia, as well as areas in eastern Kentucky and southwestern Virginia, were combined and divided into two equal datasets. These datasets served as validation areas at the end of each training

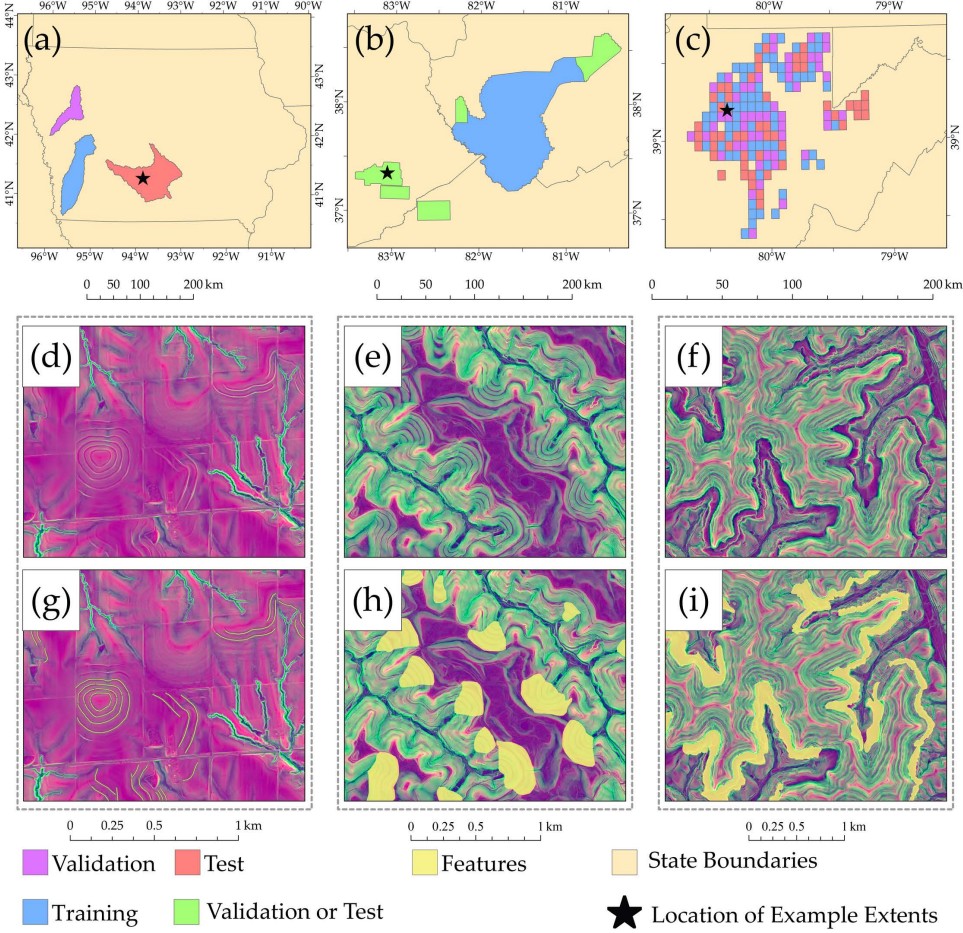

**Fig 7. Training, test, and validation areas for (a) agricultural terraces (terraceDL) dataset [ 29] in Iowa, USA; (b) surface coal mining valley fill faces (vfillDL) dataset [30] in southern West Virginia, eastern Kentucky, and southwestern Virginia, USA; and (c) historic mine benches (minebenchDL) dataset [31] in northern West Virginia, USA.** (d) through (i) show example terrain surfaces and associated geomorphic features as examples.

epoch and as the test set for final model evaluation. Collectively, these regions contained 1,178 valley fills. We specifically focused on mapping valley fill faces.

The minebenchDL dataset [31] (Figs 7(c), (f), and (i)) was produced by the authors through on-screen digitizing and interpretation of topographic data derived from lidar, aerial orthophotography, and ancillary mine disturbance extents. Mine benches are remnants of historic surface coal mining. Prior to SMCRA, operators were not required to reclaim the landscape to "approximate original contour"; as a result, remnant benches are abundant in the Appalachian coalfields. During the mining process, coal seams were excavated along topographic contour, resulting in long, linear highwalls and associated benches that meander along topographic contour in areas of disturbance. Overburden rock material was often placed on the slopes beneath the benches, further modifying the terrain. Many of the mine sites have now been revegetated and are not obvious in aerial imagery. However, the benches are generally very distinctive in high spatial resolution digital terrain data. As highlighted in Fig 7(c), the data were partitioned into separate training, validation, and test sets based on 1:24,000 scale United States Geological Survey (USGS) quarter quadrangle boundaries. The 169 mapped quarter quadrangles were randomly partitioned such that 69 were used for training, 50 were used for validation, and 50

were used for testing. Only quarter quads that included mine bench features were used. Collectively, 2,416 mine benches occurred within the mapped extent.

The characteristics of each dataset have important implications for model performance and interpretation of results. Terrace and mine bench features are typically long and narrow, resulting in a higher proportion of edge pixels within image chips and introducing challenges for segmentation, particularly in boundary delineation. In contrast, valley fill features tend to be larger and more compact, with greater interior area and fewer edge pixels. Additionally, the number of available training chips varied considerably across datasets (see Table 1), which may have influenced training dynamics, especially in experiments using limited sample sizes. These differences underscore the importance of considering both the spatial structure of features and the quantity of training data when interpreting segmentation outcomes and selecting appropriate UNet configurations.

### Land surface parameters (LSPs)

Variables derived from DTMs, termed LSPs, were used to characterize the terrain surface and serve as predictor variables. We specifically used a three-layer composite originally proposed by Drs. Odom and Doctor of the USGS. Maxwell et al. [79] highlighted that this approach improves the accuracy of geomorphic feature extraction in comparison to more traditional terrain representations (e.g., hillshades and slopeshades). The model's three layers consist of: a topographic position index (TPI) with a 50 m radius moving window to characterize hillslope positions, the square root of slope to measure steepness, and a second TPI with a more localized annulus moving window capturing detailed relief and roughness. These layers were produced using R [28] and the terra [80] and MultiscaleDTM [81] packages. The resulting feature space is shown as the terrain images in Fig 7(d) through (i) above.

### Image chip generation and data partitions

It is necessary to train CNNs using rectangular image chips rather than individual pixels since spatial context is modeled with learned kernels [82–84]. In this study, LSP raster grids and their associated pixel-level labels within the defined training, validation, and test extents were divided into non-overlapping 512×512 cell chips. Table 1 summarizes the number of training, validation, and test chips used for each dataset. To investigate the impact of training sample size, random subsets of the three training datasets were generated with sample sizes of 50, 100, 250, and 500 chips, in addition to using all available training chips. Notably, only chips containing at least one pixel mapped to the positive class were included. These chips were created using geodl [27].

### Training process

All models were trained using the geodl [27] R [28] package. The package offers a DL semantic segmentation implementation in R [85] and C++ without needing a Python/PyTorch environment [86,87]. It makes use of the torch [88], luz [89], and terra [80] packages. For all configurations, encoder blocks 1–4 generated 16, 32, 64, and 128 feature maps, respectively, while the bottleneck block produced 256 feature maps. Decoder blocks 1–4 generated 128, 64, 32, and 16 feature maps, respectively. Table 2 describes the different model configurations used and lists their associated number of trainable parameters.

**Table 1. Overview of the available training, validation, and test chips for each dataset used.**

| Dataset | Terraces | Valley Fills | Mine Benches |
|---|---|---|---|
| **Training** | 50, 100, 250, 500, 3981 | 50, 100, 250, 500, 1222 | 50, 100, 250, 500, 4164 |
| **Validation** | 1488 | 423 | 2334 |
| **Test** | 4183 | 423 | 2304 |

**Table 2. Summary of UNet-based models with descriptions of each configuration.**

| Model/Augmentation Name | Description |
|---|---|
| Base UNet | Baseline UNet model without additional modules |
| Leaky ReLU Activation | UNet using leaky ReLU as the activation functions (negative slope term = 0.01) |
| Swish Activation | UNet using swish as the activation functions |
| Residual Blocks | UNet with residual connections in all encoder/decoder blocks |
| DC Bottleneck | UNet with DC module used as bottleneck block (dilation rates of 1, 2, 4, 8, and 16 were used and 16 feature maps were generated at each dilation rate; these were then processed into 256 feature maps using 1 × 1 2D convolution) |
| Squeeze and Excitation | UNet with SE modules added (reduction term = 8) |
| Attention Gates | UNet with AG modules added |
| MobileUNet | UNet with MobileNetV2 as the encoder backbone |

For all experiments, models were trained for 25 epochs, or passes over the entire training set, using a mini-batch size of 10 and the AdamW optimizer [90,91]. A learning rate of 1e-3 was used for all trainable components of the model. All models were initialized using random weights, and no pre-training or transfer learning was applied. To mitigate overfitting, random horizontal and/or vertical flips were applied to the training chips with a probability of 0.5.

The geodl package provides a modified implementation of the unified focal loss framework [92], allowing for defining various loss metrics with a consistent implementation and parameter set by adjusting lambda ($\lambda$), gamma ($\gamma$), delta ($\delta$), and class weight terms [27]. Given the significant class imbalance in the datasets, where only a small proportion of cells were classified as positive, or as the geomorphic feature of interest, a focal Tversky loss [62] was used with $\lambda = 0$, $\gamma = 0.8$, and $\delta = 0.65$. $\lambda$ controls the relative weight of distribution- and region-based losses and was set to 0 to consider only the region-based loss. $\gamma$, which manages the application of focal loss and assigns greater weight to difficult-to-predict classes [62,93–95], was set to 0.8. A value of 1 applies no focal correction while values less than 1 but greater than 0 apply a focal correction, with lower values placing more weight on difficult-to-predict classes or samples. $\delta$, which controls the relative weight of false positive (FP) and false negative (FN) errors for each class, was set to 0.65 to place a higher weight on FP errors compared to FN errors for each class [92].

To compare models beyond accuracy, we considered computational complexity and model efficiency using standardized metrics: the number of floating-point operations (FLOPs), multiply-add operations (MACs), and total number of learnable parameters, as shown in Table 3. These measures offer architecture-specific insights into scalability and feasibility for practical deployment. While training and inference times are important in applied contexts, they were not used for direct comparison due to their sensitivity to implementation-specific variables such as hardware specifications, mini-batch size, and optimizer settings. By focusing on FLOPs, MACs, and parameter count, we provide a more controlled and reproducible basis for evaluating the computational efficiency of different UNet configurations for large-scale geomorphic feature mapping.

## Model assessment

As opposed to using the model state after the final training epoch as the final model, the model state produced by the training epoch that provided the highest F1-score for the validation data was selected. Once the final model was chosen for each tested configuration and dataset combination, each model was applied to the withheld testing data for its associated problem.

Using the predictions and associated pixel-level labels for the test set, performance was evaluated with the assessment metrics described in Table 4, including overall accuracy, recall, precision, and F1-score. Overall accuracy represents the proportion of correctly classified pixels relative to the total number of pixels. Precision and recall offer insights into the

**Table 3. Model complexity and computational cost comparison. 1 GFLOP = 1 billion FLOPs; 1 GMAC = 1 billion MACs.**

| Model/Augmentation Name | Trainable Parameters | FLOPs (GFLOPs) | MACs (GMACs) |
|---|---|---|---|
| Base UNet | 2,315,186 | 303.58 | 150.87 |
| Leaky ReLU Activation | 2,315,186 | 303.58 | 150.87 |
| Swish Activation | 2,315,186 | 303.58 | 150.87 |
| Residual Blocks | 2,426,242 | 323.26 | 160.39 |
| DC Bottleneck | 1,542,562 | 287.76 | 142.96 |
| Squeeze and Excitation | 2,320,626 | 303.66 | 150.87 |
| Attention Gates | 2,382,158 | 307.79 | 152.9 |
| MobileUNet | 6,459,578 | 307.07 | 151.92 |

**Table 4. Assessment metrics used in study. TP = true positive; TN = true negative; FP = false positive; FN = false negative.**

| Metric | Overall Accuracy | Recall | Precision | F1-Score |
|---|---|---|---|---|
| Equation | $\dfrac{TP+TN}{TP+TN+FP+FN}$ | $\dfrac{TP}{TP+FN}$ | $\dfrac{TP}{TP+FP}$ | $\dfrac{2 \times Recall \times\ Precision}{Recall\ +\ Precision}$ |

model's ability to minimize commission and omission errors for the positive class, respectively. Relative to the positive case, or the geomorphic feature of interest, commission errors result from background pixels being incorrectly predicted as the geomorphic feature of interest while omission errors result from pixels occurring within the geomorphic feature of interest being incorrectly predicted to the background class. The F1-score, being the harmonic mean of precision and recall, provides a combined measure of the model's performance [39,93,96,97].

## Results

### Performance using the entire training set

In this section, we focus on the results for the eight different UNet-based models trained using the full training dataset. Fig 8 provides the training loss curves for the mineBenchDL, vfillDL, and terraceDL experiments. Generally, the training loss stabilized at different losses across the datasets. The vfillDL dataset achieved the lowest training loss, followed by the terraceDL dataset. The mineBenchDL dataset had the highest training loss among the three, indicating greater difficulty in optimizing the models for this feature extraction task. For all three datasets, the MobileUNet, Residual Blocks, and DC Bottleneck configurations were particularly effective, showing rapid convergence and lower final loss values. Overall, there was more variability between model configurations for the mine bench and valley fill face problems in comparison to the agricultural terrace problem.

### Impact of sample size

Figs 9 and 10 illustrate the training loss and validation F1-score, respectively, for the mineBenchDL, terraceDL, and vfillDL datasets across 25 training epochs. These figures reflect all training sample sizes, 50, 100, 250, 500, and the full set of 512 × 512 cell chips across various model configurations. As anticipated, increasing training sample sizes generally enhanced model performance, with differences among model configurations diminishing as sample sizes reached 500 chips. However, the architecture design exerted a more pronounced influence when models were trained on smaller sample sizes. Specifically, the Residual Blocks, Swish Activation, and DC Bottleneck configurations consistently achieved superior performance with reduced sample sizes. Models employing MobileNetv2 as the encoder faced performance challenges at lower sample sizes, likely due to the increased complexity of this architecture, which may require larger training sets to optimize effectively. This trend suggests that, while model architecture plays a role

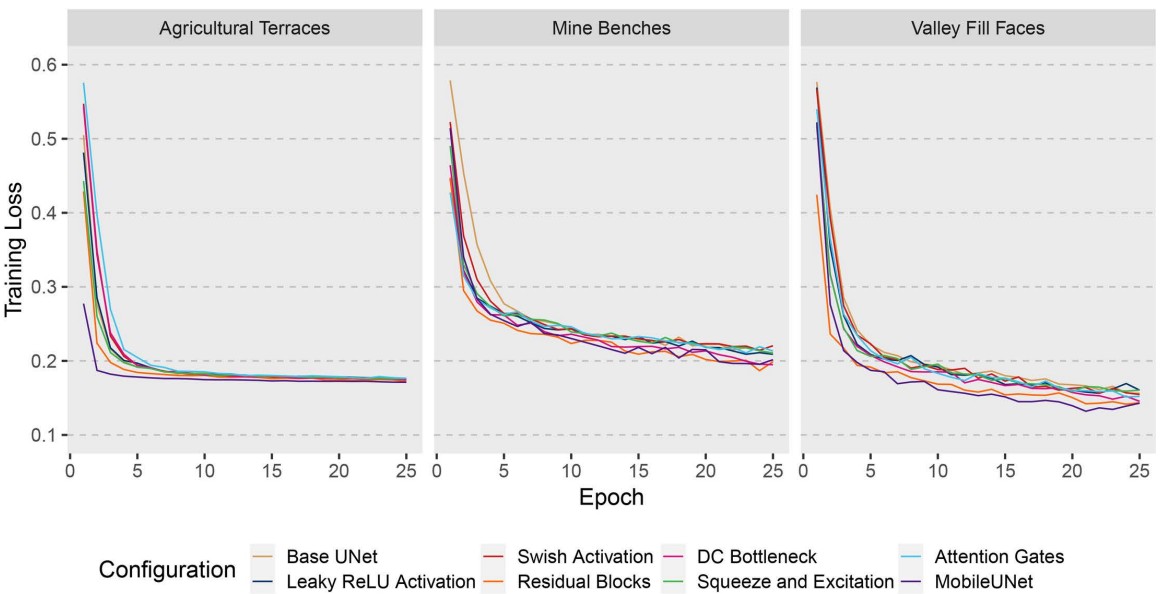

**Fig 8. Training loss for terraceDL, mineBenchDL, and vfillDL datasets using all training samples and different model configurations across 25 training epochs.**

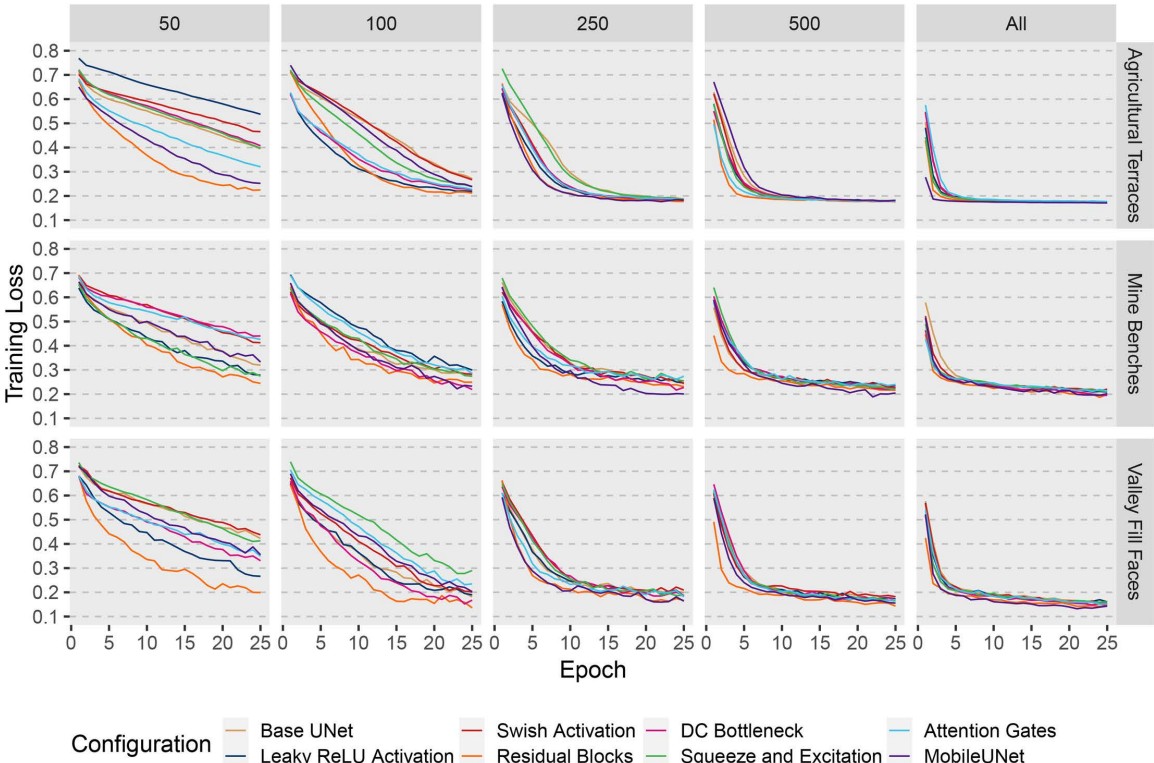

**Fig 9. Training loss for terraceDL, mineBenchDL, and vfillDL datasets using varying training sample sizes and different model configurations across 25 training epochs.**

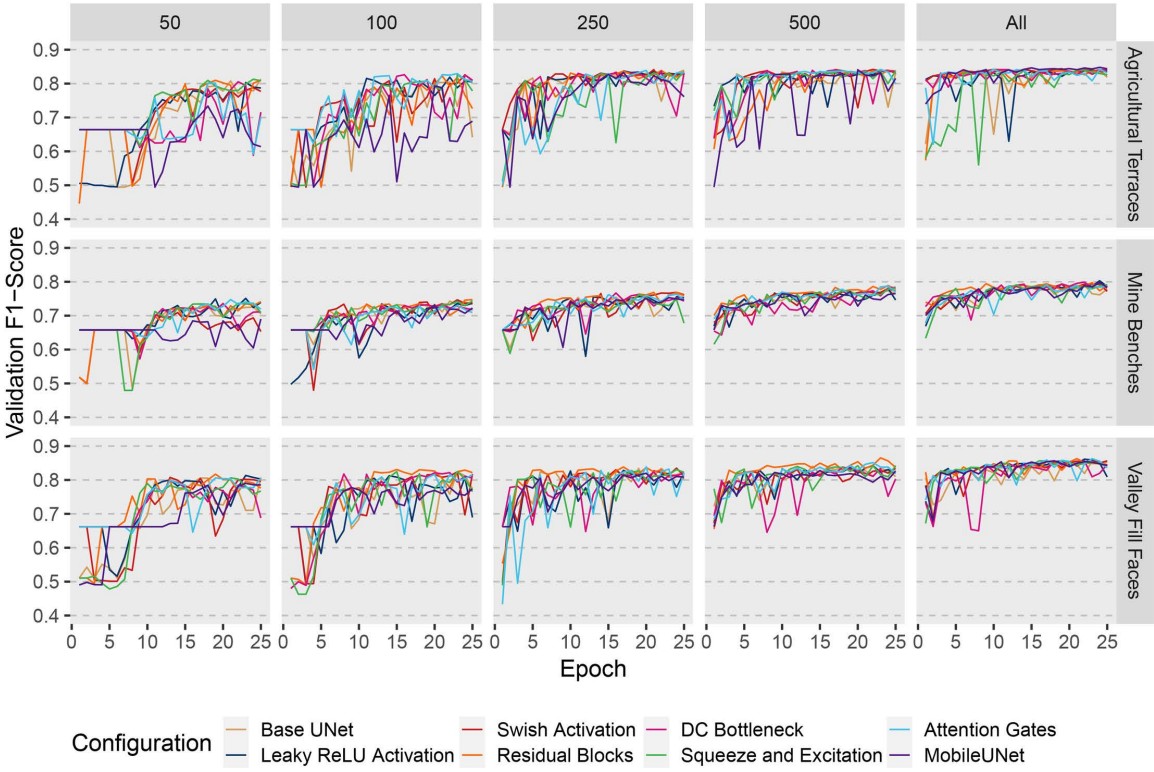

**Fig 10. Validation F1-score for terraceDL, mineBenchDL, and vfillDL datasets using varying training sample sizes and different model configurations across 25 training epochs.**

in performance, the importance of specific configurations diminishes as sample sizes increase; thus, model configuration choices may be of greater concern when only small training samples sizes are available. It should be noted that smaller sample sizes result in fewer total parameter updates when using the same mini-batch size since parameters are updated after each mini-batch is processed as opposed to after each complete pass over the training data, or epoch.

## Comparison of test set predictions

As described in the Methods section above, the model from the epoch that achieved the highest F1-score for the validation data for each configuration was selected as the final model, which was then applied to the test data to calculate the test set loss and assessment metrics. Fig 11 presents the F1-score results using the varying training set sizes.

Across datasets, increasing the sample size generally improved the F1-score, largely independent of the model configuration. However, model configuration had a more pronounced impact when using smaller sample sizes. Interestingly, the MobileNetv2-based models showed lower performance at smaller sample sizes, possibly due to the higher complexity of the architecture. As sample sizes increased, the performance differences among configurations decreased. Although differences between F1-scores were small, the Residual Blocks configuration provided the highest F1-score for the mineBenchDL and vfillDL datasets, while the Swish Activation configuration was more effective for the terraceDL dataset using sample sizes of 250 and 500 chips. When using the full dataset for the terraceDL datasets, the MobileNetv2-based model provided the highest F1-score.

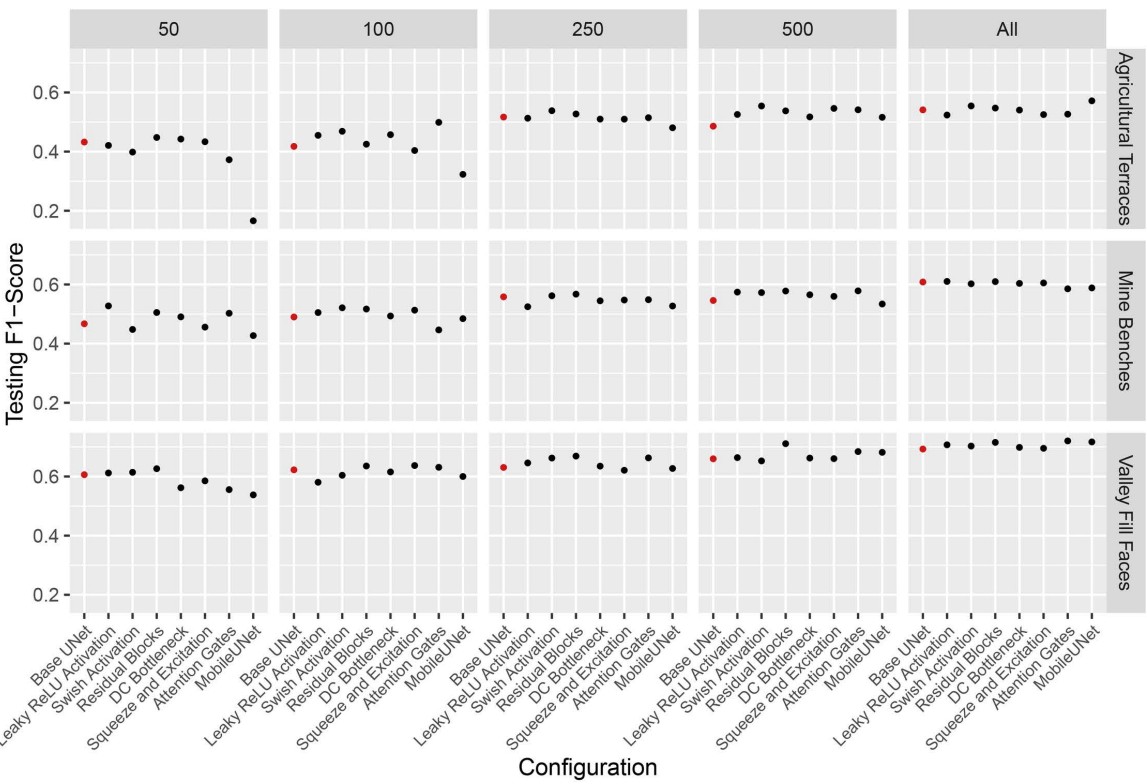

**Fig 11. F1-score calculated from the withheld test data using different architectural configurations and training set sizes.** Red points indicate the Base UNet model.

Table 5 presents the assessment metrics for the terraceDL predictions across all model configurations and sample sizes. Similarly, Table 6 summarizes the results for the mineBenchDL problem while Table 7 does the same for the vfillDL problem.

A key observation is that models tended to stabilize when provided with 500 or more training samples, as improvements in F1-score, recall, and precision became more incremental. Although overall accuracies were consistently high across all experiments, this is largely due to class imbalance. A significant portion of the landscape is classified as the background class, which is generally well predicted, resulting in a large proportion of correctly mapped pixels. Consequently, overall accuracy can inflate perceived model performance even when the positive class is poorly segmented. Therefore, it is crucial to evaluate the models using F1-score, recall, and precision specifically for the positive cases.

To visualize model performance, we applied models trained on 500 samples from the terraceDL, mineBenchDL, and vfillDL training datasets to predict spatial subsets from their respective test areas. Figs 12–14 display the resulting predictions for the terraceDL, mineBenchDL, and vfillDL testing areas, respectively. Despite similar F1-scores, as highlighted in Fig 11, visualizing the results over a subset of the test areas suggests notable differences between model configurations in regards to continuity of the features generated, overprediction or FPs, underprediction or FNs, and general sources of confusion or error. For example, the Base UNet configuration for the agricultural terraces problem (Fig 12) tended to misclassify some road segments as terraces. For the mine bench problem (Fig 13), the MobileUNet configuration had less FP occurrences in the floodplain in comparison to other model configurations. For the valley fill face problem (Fig 14), the models generally varied based on the abundance of small clusters of FP features predicted.

**Table 5. Testing set assessment metrics for prediction of agricultural terraces using different architectural configurations and varying sample sizes.**

| Configurations | Training sample size (Number of chips) | OA | Recall | Precision | F1-score |
|---|---|---|---|---|---|
| Base UNet | 50 | 0.988 | 0.418 | 0.448 | 0.432 |
| | 100 | 0.988 | 0.379 | 0.464 | 0.418 |
| | 250 | 0.991 | 0.458 | 0.593 | 0.517 |
| | 500 | 0.991 | 0.39 | 0.645 | 0.486 |
| | 3981 | 0.99 | 0.509 | 0.578 | 0.541 |
| Leaky ReLU Activation | 50 | 0.987 | 0.419 | 0.423 | 0.421 |
| | 100 | 0.988 | 0.462 | 0.448 | 0.455 |
| | 250 | 0.99 | 0.458 | 0.582 | 0.513 |
| | 500 | 0.991 | 0.468 | 0.6 | 0.526 |
| | 3981 | 0.991 | 0.454 | 0.618 | 0.524 |
| Swish Activation | 50 | 0.986 | 0.424 | 0.376 | 0.398 |
| | 100 | 0.99 | 0.411 | 0.545 | 0.469 |
| | 250 | 0.99 | 0.513 | 0.566 | 0.538 |
| | 500 | 0.991 | 0.508 | 0.61 | 0.554 |
| | 3981 | 0.991 | 0.538 | 0.572 | 0.554 |
| Residual Blocks | 50 | 0.988 | 0.441 | 0.455 | 0.448 |
| | 100 | 0.99 | 0.327 | 0.608 | 0.425 |
| | 250 | 0.99 | 0.489 | 0.571 | 0.527 |
| | 500 | 0.991 | 0.469 | 0.63 | 0.538 |
| | 3981 | 0.991 | 0.504 | 0.599 | 0.547 |
| DC Bottleneck | 50 | 0.989 | 0.396 | 0.501 | 0.442 |
| | 100 | 0.989 | 0.42 | 0.502 | 0.457 |
| | 250 | 0.99 | 0.472 | 0.554 | 0.51 |
| | 500 | 0.991 | 0.447 | 0.613 | 0.517 |
| | 3981 | 0.99 | 0.484 | 0.612 | 0.54 |
| Squeeze and Excitation | 50 | 0.985 | 0.507 | 0.379 | 0.433 |
| | 100 | 0.99 | 0.312 | 0.573 | 0.404 |
| | 250 | 0.99 | 0.494 | 0.527 | 0.51 |
| | 500 | 0.991 | 0.475 | 0.643 | 0.546 |
| | 3981 | 0.991 | 0.465 | 0.604 | 0.525 |
| Attention Gates | 50 | 0.99 | 0.282 | 0.548 | 0.373 |
| | 100 | 0.99 | 0.435 | 0.585 | 0.499 |
| | 250 | 0.991 | 0.437 | 0.627 | 0.515 |
| | 500 | 0.991 | 0.497 | 0.595 | 0.541 |
| | 3981 | 0.99 | 0.461 | 0.614 | 0.527 |
| MobileUNet | 50 | 0.989 | 0.098 | 0.535 | 0.166 |
| | 100 | 0.99 | 0.216 | 0.641 | 0.323 |
| | 250 | 0.99 | 0.425 | 0.552 | 0.481 |
| | 500 | 0.991 | 0.456 | 0.595 | 0.516 |
| | 3,981 | 0.992 | 0.512 | 0.647 | 0.572 |

**Table 6. Testing set assessment metrics for prediction of historic mine benches using different architectural configurations and varying sample sizes.**

| Configurations | Training sample size (Number of chips) | OA | Recall | Precision | F1-score |
|---|---|---|---|---|---|
| Base UNet | 50 | 0.927 | 0.536 | 0.414 | 0.467 |
| | 100 | 0.932 | 0.545 | 0.446 | 0.49 |
| | 250 | 0.952 | 0.513 | 0.612 | 0.558 |
| | 500 | 0.947 | 0.53 | 0.563 | 0.546 |
| | 4164 | 0.954 | 0.594 | 0.623 | 0.608 |
| Leaky ReLU Activation | 50 | 0.942 | 0.541 | 0.515 | 0.528 |
| | 100 | 0.938 | 0.531 | 0.482 | 0.505 |
| | 250 | 0.949 | 0.47 | 0.593 | 0.525 |
| | 500 | 0.946 | 0.611 | 0.541 | 0.574 |
| | 4164 | 0.951 | 0.643 | 0.581 | 0.61 |
| Swish Activation | 50 | 0.917 | 0.562 | 0.372 | 0.448 |
| | 100 | 0.941 | 0.537 | 0.506 | 0.521 |
| | 250 | 0.951 | 0.527 | 0.602 | 0.562 |
| | 500 | 0.953 | 0.524 | 0.632 | 0.573 |
| | 4164 | 0.959 | 0.523 | 0.71 | 0.602 |
| Residual Blocks | 50 | 0.934 | 0.566 | 0.456 | 0.505 |
| | 100 | 0.938 | 0.557 | 0.483 | 0.517 |
| | 250 | 0.951 | 0.537 | 0.601 | 0.567 |
| | 500 | 0.946 | 0.617 | 0.543 | 0.578 |
| | 4164 | 0.958 | 0.551 | 0.683 | 0.61 |
| DC Bottleneck | 50 | 0.939 | 0.493 | 0.488 | 0.491 |
| | 100 | 0.936 | 0.524 | 0.467 | 0.493 |
| | 250 | 0.94 | 0.605 | 0.495 | 0.545 |
| | 500 | 0.95 | 0.545 | 0.587 | 0.565 |
| | 4164 | 0.954 | 0.591 | 0.616 | 0.604 |
| Squeeze and Excitation | 50 | 0.914 | 0.604 | 0.366 | 0.456 |
| | 100 | 0.943 | 0.501 | 0.526 | 0.513 |
| | 250 | 0.946 | 0.549 | 0.546 | 0.547 |
| | 500 | 0.945 | 0.583 | 0.539 | 0.56 |
| | 4164 | 0.958 | 0.534 | 0.698 | 0.605 |
| Attention Gates | 50 | 0.942 | 0.495 | 0.511 | 0.503 |
| | 100 | 0.915 | 0.575 | 0.365 | 0.446 |
| | 250 | 0.945 | 0.565 | 0.533 | 0.549 |
| | 500 | 0.947 | 0.606 | 0.553 | 0.578 |
| | 4164 | 0.953 | 0.556 | 0.618 | 0.585 |
| MobileUNet | 50 | 0.922 | 0.486 | 0.381 | 0.427 |
| | 100 | 0.939 | 0.483 | 0.486 | 0.485 |
| | 250 | 0.954 | 0.426 | 0.691 | 0.527 |
| | 500 | 0.939 | 0.585 | 0.491 | 0.534 |
| | 4164 | 0.952 | 0.577 | 0.6 | 0.588 |

**Table 7. Testing set assessment metrics for prediction of surface coal mine valley fill faces using different architectural configurations and varying sample sizes.**

| Configurations | Training sample size (Number of chips) | OA | Recall | Precision | F1-score |
|---|---|---|---|---|---|
| Base UNet | 50 | 0.971 | 0.548 | 0.678 | 0.606 |
| | 100 | 0.973 | 0.55 | 0.717 | 0.623 |
| | 250 | 0.975 | 0.523 | 0.794 | 0.631 |
| | 500 | 0.975 | 0.589 | 0.75 | 0.66 |
| | 1222 | 0.978 | 0.617 | 0.79 | 0.693 |
| Leaky ReLU Activation | 50 | 0.971 | 0.566 | 0.667 | 0.612 |
| | 100 | 0.973 | 0.466 | 0.77 | 0.581 |
| | 250 | 0.972 | 0.618 | 0.676 | 0.646 |
| | 500 | 0.975 | 0.608 | 0.73 | 0.664 |
| | 1222 | 0.978 | 0.649 | 0.777 | 0.707 |
| Swish Activation | 50 | 0.969 | 0.609 | 0.62 | 0.614 |
| | 100 | 0.974 | 0.485 | 0.801 | 0.604 |
| | 250 | 0.972 | 0.662 | 0.663 | 0.662 |
| | 500 | 0.974 | 0.598 | 0.719 | 0.653 |
| | 1222 | 0.979 | 0.622 | 0.809 | 0.703 |
| Residual Blocks | 50 | 0.972 | 0.585 | 0.675 | 0.626 |
| | 100 | 0.975 | 0.543 | 0.766 | 0.636 |
| | 250 | 0.974 | 0.638 | 0.704 | 0.669 |
| | 500 | 0.979 | 0.623 | 0.828 | 0.711 |
| | 1222 | 0.978 | 0.67 | 0.768 | 0.715 |
| DC Bottleneck | 50 | 0.97 | 0.478 | 0.682 | 0.562 |
| | 100 | 0.974 | 0.507 | 0.782 | 0.615 |
| | 250 | 0.972 | 0.608 | 0.665 | 0.635 |
| | 500 | 0.975 | 0.595 | 0.747 | 0.662 |
| | 1222 | 0.978 | 0.618 | 0.803 | 0.698 |
| Squeeze and Excitation | 50 | 0.971 | 0.498 | 0.71 | 0.585 |
| | 100 | 0.971 | 0.624 | 0.651 | 0.637 |
| | 250 | 0.976 | 0.492 | 0.842 | 0.621 |
| | 500 | 0.976 | 0.566 | 0.792 | 0.66 |
| | 1222 | 0.978 | 0.622 | 0.789 | 0.695 |
| Attention Gates | 50 | 0.972 | 0.425 | 0.803 | 0.556 |
| | 100 | 0.969 | 0.652 | 0.612 | 0.631 |
| | 250 | 0.975 | 0.6 | 0.741 | 0.663 |
| | 500 | 0.976 | 0.625 | 0.756 | 0.684 |
| | 1222 | 0.978 | 0.709 | 0.732 | 0.72 |
| | 50 | 0.971 | 0.403 | 0.809 | 0.538 |
| | 100 | 0.971 | 0.523 | 0.704 | 0.6 |
| MobileUNet | 250 | 0.974 | 0.511 | 0.813 | 0.627 |
| | 500 | 0.975 | 0.625 | 0.75 | 0.682 |
| | 1222 | 0.978 | 0.648 | 0.803 | 0.717 |

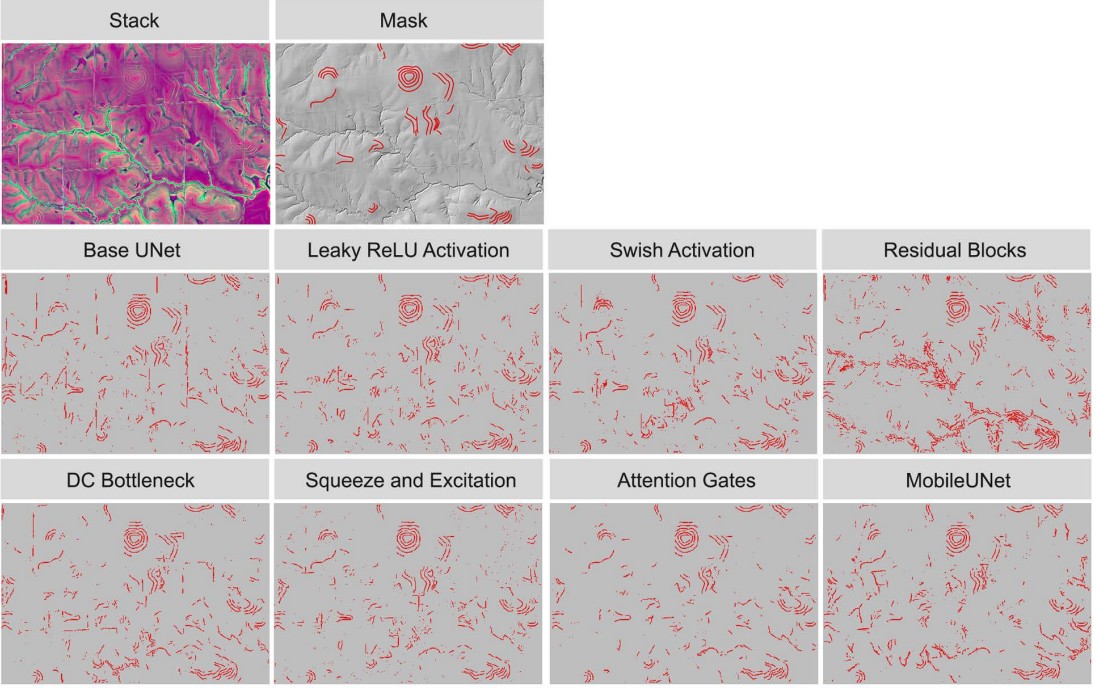

**Fig 12. Comparison of agricultural terraces detection results using different architectural configurations.**

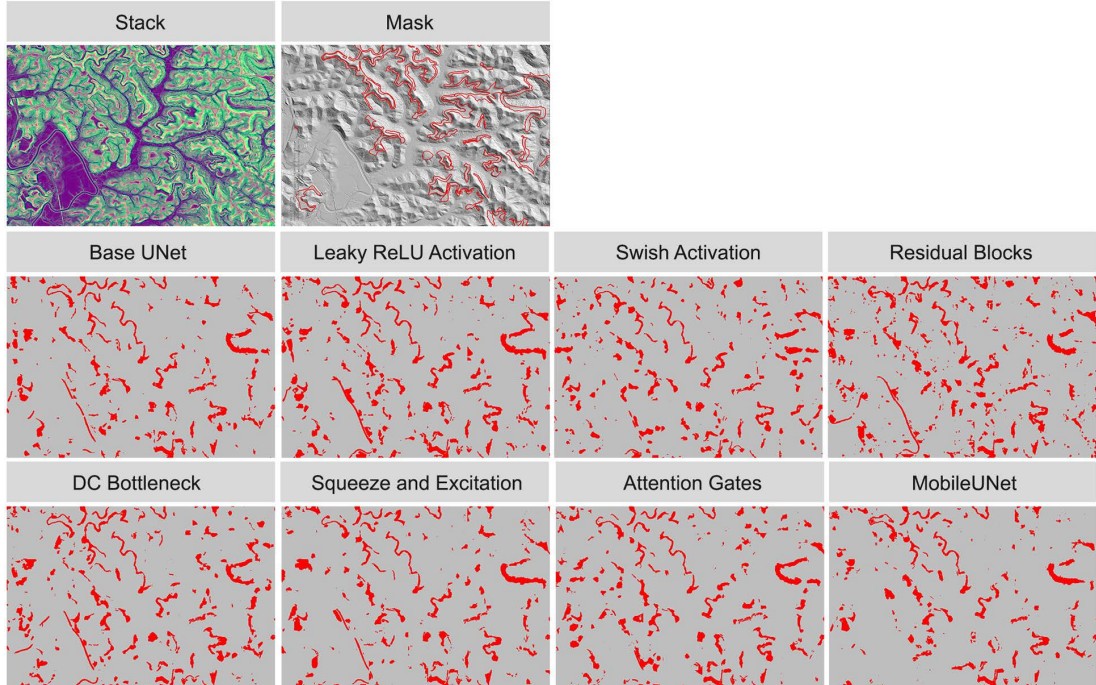

**Fig 13. Comparison of historic mine benches detection results using different architectural configurations.**

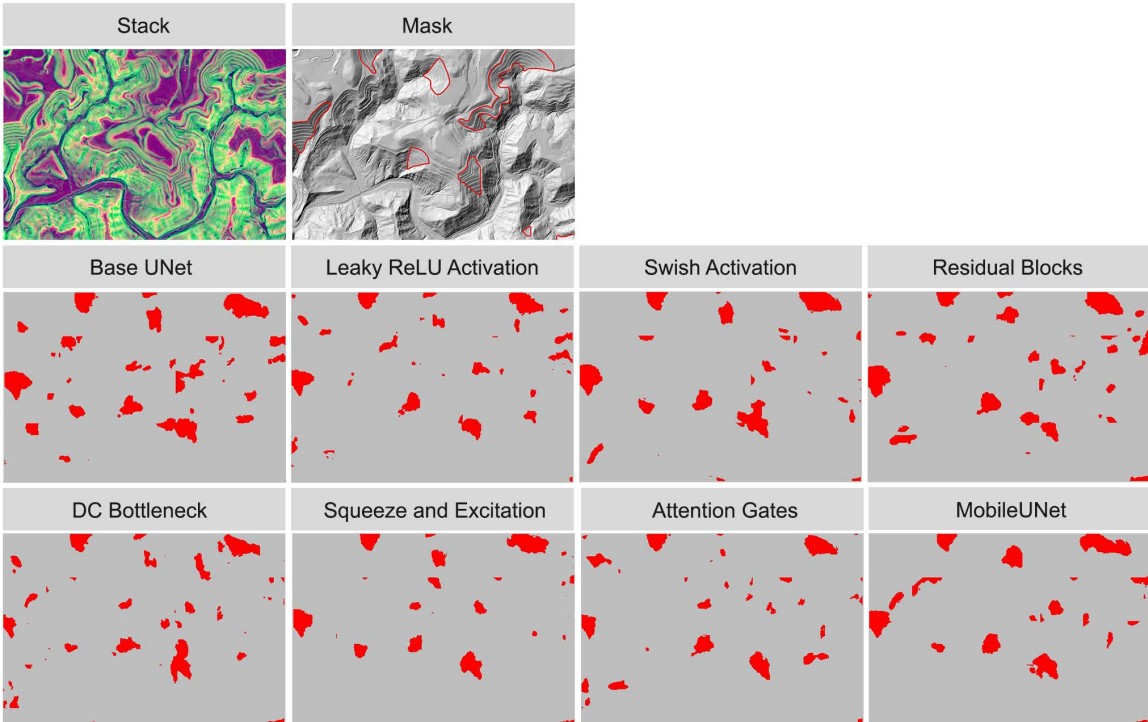

**Fig 14.  Comparison of valley fill faces detection results using different architectural configurations.**

Generally, this highlights the value of visualizing results over new extents as opposed to only relying on assessment metrics to quantify model performance. Differences between models that may be important to the production team or end users may not be well reflected in assessment metrics. For example, if the products will be manually edited, it may be desirable to have more issues of FPs as opposed to FNs since it may be more time consuming to digitized missed objects as opposed to deleting false detections. The contiguity of generated features may also be of importance, especially if the data will be converted to vector objects or the production team plan to manually edit generated features to improve the final product.

## Bootstrap analysis of model performance

To further assess model robustness, we used models trained with 500 samples from the terraceDL, mineBenchDL, and vfillDL training datasets to re-calculate assessment metrics based on 10 bootstrap samples, each consisting of a random selection of 300 samples from the available test chips. This analysis, presented in Fig 15, displays the overall accuracy, F1-score, precision, and recall for each configuration across the mineBenchDL, terraceDL, and vfillDL datasets. The error bars in the figure represent the range of values obtained from the bootstrap samples, providing insight into the variability of each metric across different configurations and datasets.

The results generally suggest fairly stable performance for predicting subsets of the test set, as indicated by generally small differences in resulting metrics between the bootstrap samples. Further, there is generally overlap between different model configurations, which further highlights that the differences in performance between architectural configurations were limited once sample sizes reached 500 training chips. Overall accuracy was generally higher and less variable than the other metrics, which we attribute to class imbalance, or the large number of background cells in comparison to cells

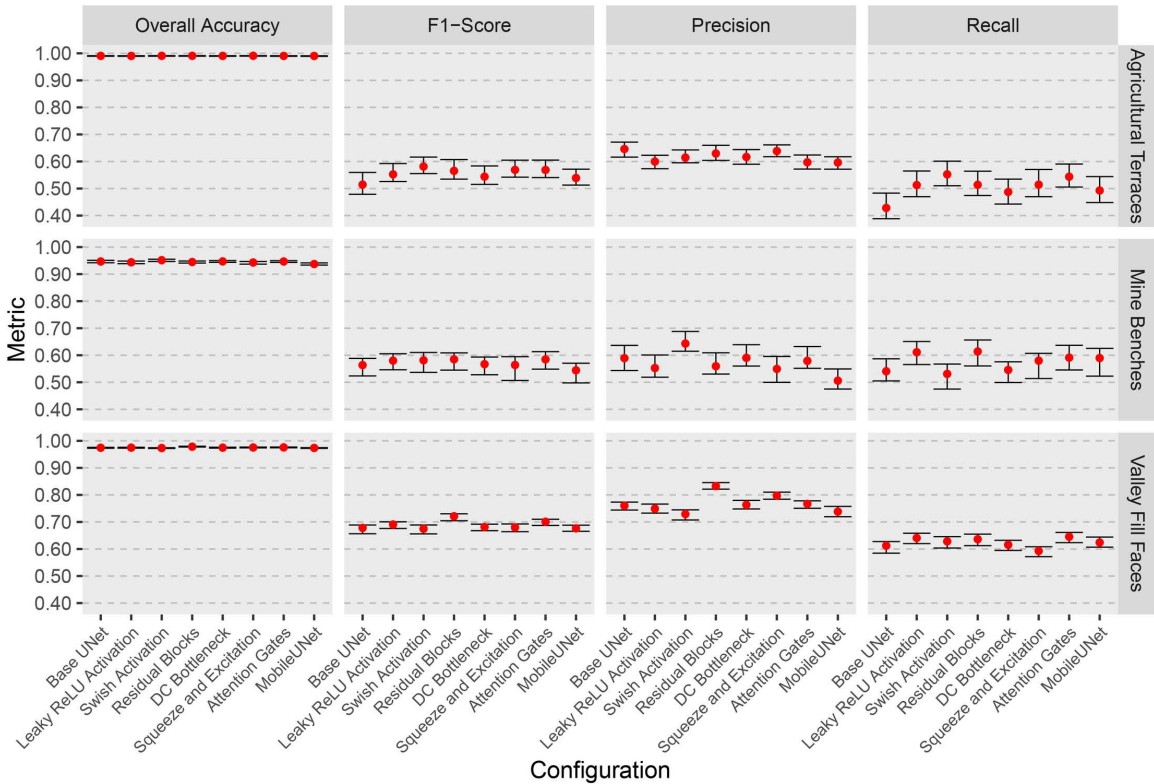

**Fig 15. Bootstrap analysis of overall accuracy, F1-score, precision, and recall across model configurations for mineBenchDL, terraceDL and vfillDL datasets.** Error bars represent variability from 10 bootstrap test samples, highlighting performance stability differences across configurations.

mapped to the feature of interest. Despite some overlapping ranges, some architectural configurations stand out, such as residual blocks for the valley fill face problem.

## Discussion

Several critical findings emerge from this work, offering valuable insights into the performance and optimization of UNet-based architectures for anthropogenic geomorphic feature extraction:

1. Architectural configuration is generally more important when only small training set sizes (e.g., smaller than 500 samples), are available. When sample sizes are larger, base UNet performs well, and performance gains from modifications or added modules are marginal.

2. Certain modules stand out as useful: residual connections, swish activation, and using dilated convolution in the bottleneck.

3. Complex backbone encoders, such as MobileNetV2, may require larger training sets to be successfully implemented due to their use of depthwise separable convolutions and linear bottlenecks, which demand more data to effectively learn nuanced spatial features particularly without pretraining. Larger architectures may also be more prone to overfitting, especially when the training set size is small and transfer learning is not implemented. Transfer learning was not explored in this study; however, it was assessed in our prior study, Maxwell et al. [98], where transfer learning was generally noted to be of value for geomorphic feature extraction tasks.

4. Assessment metrics may not reflect all important considerations, such as the visual continuity of features. This highlights the importance of visualizing output predictions over new geographic extents and not solely relying on assessment metrics.

In addition to segmentation performance, computational complexity is an important factor in evaluating the practicality of deep learning architectures for large-scale or resource-constrained applications. Table 3 presents a comparison of model complexity and computational cost across the UNet variants evaluated in this study. The number of trainable parameters varied considerably among models, ranging from 1.54 million for the DC Bottleneck model to 6.46 million for MobileUNet. In terms of computational cost, measured by FLOPs and MACs, the DC Bottleneck model was the most efficient, with the lowest FLOPs (287.76 GFLOPs) and MACs (142.96 GMACs). The DC Bottleneck variant is especially noteworthy for being both lightweight and computationally efficient while still performing competitively. Its lower resource demands make it particularly attractive for deployment in scenarios with limited computing resources. Conversely, the MobileUNet configuration exhibited the highest parameter count, nearly three times that of the base UNet, but with only modest increases in computational operations. This highlights MobileNetV2's depthwise separable convolutions, which effectively trade increased memory usage for more efficient computation. Additionally, while the residual blocks and attention modules introduced moderate increases in computational complexity, their use might still be justified if they lead to substantial improvements in model accuracy or feature delineation. Therefore, we emphasize that decisions about adopting complex modules should balance measurable performance gains against computational costs and resource availability.

Visual assessments of the LSPs and hillshades used in this study reveal that agricultural features and mine benches stand out more distinctly from the landscape background compared to valley fill faces, which tend to have more gradational boundaries, particularly at the top of the fill where they meet the reclaimed mine complex. In our previous work [79], we documented that the DL models generally performed better on the valley fill face problem than on the terrace problem. This discrepancy was largely attributed to the unique shape of the terraces, which are typically long and narrow, leading to a higher proportion of edge cells. Similarly, mine benches, which are also long and narrow, following the original topographic contours, have a similar issue with a greater proportion of edge cells. In contrast, valley fill faces are generally larger and less elongated, resulting in a lower proportion of edge cells due to their substantial interior area. Therefore, the lower performance on terrace and mine bench problems compared to valley fill faces can be at least partially attributed to the shape of these features rather than the inherent difficulty in distinguishing them from the landscape background. This underscores the importance of considering factors beyond the complexity of the mapping problem when interpreting assessment metrics.

Our findings indicate that at least 500 training chips should be used for these geomorphic feature extraction problems. However, this threshold may not apply universally due to varying levels of difficulty resulting from distinguishing features of interest from the landscape, variability in feature presentation, and issues of fuzzy or gradational boundaries. For all three problems, model performance generally stabilized with 500 or more training chips, suggesting that collecting an extremely large dataset and generating thousands of image chips may not be necessary or worth the time and cost. The observed stabilization of model performance beyond 500 training chips likely results from the combined effect of dataset characteristics, particularly the degree of visual separability of the features and their consistency, and the inherent capacity of the UNet architecture, which appears sufficient to capture key spatial patterns and generalize effectively beyond this training size. However, it is important to note that this may not extrapolate to other geomorphic feature extraction tasks due to feature characteristics, number of classes being differentiated, within class heterogeneity, and the degree of differentiation between the feature(s) of interest and the rest of the landscape. Given that the same mini-batch size was used across experiments, smaller sample sizes resulted in fewer parameter updates per training epoch. This implies that using a larger number of epochs for small sample sizes (e.g., 50 or 100 chips) could be beneficial, though it might also increase the risk of overfitting [36]. Additionally, greater variability was observed between models with different configurations when sample sizes were smaller, indicating that choosing an architectural configuration is more critical with limited data.

Similar to the findings of Kugelman et al. [8], who examined different UNet configurations for retinal layer segmentation from optical coherence tomography (OCT) data, we observed that most of the UNet variations we tested showed minimal differences in performance, particularly when using larger training sample sizes. However, these differences were more pronounced in our context, possibly due to the complexity of the mapping tasks explored. Kugelman et al. [8] obtained F1-scores above 0.97 for all architectural configurations tested, which suggests that their investigated task was simpler than the three tasks explored here. They concluded that simpler architectures, such as base UNet, are sufficient, and additional modules provide only slight improvements at the cost of increased computational complexity. Our results demonstrate that while modules or modifications, such as Residual Blocks, Swish activation, and using a DC module-based bottleneck block, offer marginal gains for specific datasets, especially with reduced training set sample size, the base UNet configuration still performed adequately for geomorphic feature extraction, similar to the findings of Kugelman et al. [8]. This suggests that simpler architectures can achieve comparable performance without added complexity with adequate training set sizes, making them more efficient for practical applications when computational resources are limited.

We generally suggest incorporating architectural configurations that do not greatly increase the number of trainable parameters and computational time, such as using alternative activation functions or incorporating residual connections. It is also notable that, as configured, the DC module used in this study drastically decreased the number of trainable parameters in comparison to Base UNet since less feature maps were generated in comparison to the default bottleneck layer. This highlights that reductions in the number of feature maps or trainable parameters may not have a large negative impact on model performance. We suggest that further investigation of the impact of the number of feature maps generated at different stages in the UNet architecture is merited. Overall, while the study demonstrates the efficacy of advanced UNet-based models for anthropogenic geomorphic feature extraction, it also underscores the importance of selecting appropriate model configurations based on the specific characteristics of the dataset. Lastly, we argue that this study and prior studies, such as Kugelman et al [8], support the conceptualization of UNet as a general architecture design that can be easily reconfigured, refined, and modified with added modules. This is a positive characteristic of the architecture that is not commonly highlighted.

## Conclusion

This study demonstrates the efficacy of UNet-based architectures, augmented with advanced modules, in extracting complex anthropogenic geomorphic features across diverse landscapes. The results highlight that while architectural modifications such as Residual Blocks, Swish Activation, and DC module-based bottleneck contribute to enhanced segmentation performance, their effectiveness varies by feature type and sample size. Residual Blocks and Swish Activation notably improved performance in complex datasets like vfillDL and mineBenchDL, showcasing their adaptability to intricate landscape structures. However, performance gains plateau as training sample sizes increase, underscoring the diminishing returns of architectural complexity in large datasets. For practical applications, these findings suggest that simpler architectures may suffice when data volume is substantial, as these configurations are computationally efficient without significant accuracy loss. This adaptability is especially valuable for resource-constrained applications.

Additionally, the shape and boundary characteristics of geomorphic features, such as the elongated nature of terraces and mine benches, impact model performance, pointing to the need for feature-specific adjustments in segmentation workflows. In conclusion, this research underscores the impact of sample size on model performance, particularly when data are limited, as decisions regarding architectural configuration become increasingly critical with smaller datasets. This research contributes valuable insights into the optimization of UNet-based models for geomorphic feature segmentation and provides a foundation for future work aimed at improving the accuracy and efficiency of deep learning approaches in geospatial applications.

## Author contributions

**Conceptualization:** Sarah Farhadpour, Aaron E. Maxwell.

**Formal analysis:** Sarah Farhadpour.

**Funding acquisition:** Aaron E. Maxwell.

**Investigation:** Sarah Farhadpour.

**Methodology:** Sarah Farhadpour, Aaron E. Maxwell.

**Project administration:** Aaron E. Maxwell.

**Supervision:** Aaron E. Maxwell.

**Validation:** Sarah Farhadpour.

**Visualization:** Sarah Farhadpour.

**Writing – original draft:** Sarah Farhadpour.

**Writing – review & editing:** Sarah Farhadpour, Aaron E. Maxwell.

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
