## [Decision Letter · Decision Letter 0]

Dear Dr. Maxwell,

Thank you for submitting your manuscript to PLOS ONE. After careful consideration, we feel that it has merit but does not fully meet PLOS ONE’s publication criteria as it currently stands. Therefore, we invite you to submit a revised version of the manuscript that addresses the points raised during the review process.

We look forward to receiving your revised manuscript.

Kind regards,

Xiaoyong Sun

Academic Editor

PLOS ONE

Journal Requirements:

Funding was provided by the National Science Foundation (Federal Award ID No. 2046059: “CAREER: Mapping Anthropocene Geomorphology with Deep Learning, Big Data Spatial Analytics, and LiDAR”). Any opinions, �ndings, and conclusions or recommendations expressed in this material are those of the author(s) and do not necessarily re�ect the views of the National Science Foundation.

Reviewers' comments:

Reviewer's Responses to Questions

**Comments to the Author**

1. Is the manuscript technically sound, and do the data support the conclusions?

Reviewer #1: Yes

Reviewer #2: Yes

2. Has the statistical analysis been performed appropriately and rigorously?

Reviewer #1: Yes

Reviewer #2: Yes

3. Have the authors made all data underlying the findings in their manuscript fully available?

Reviewer #1: No

Reviewer #2: Yes

4. Is the manuscript presented in an intelligible fashion and written in standard English?

Reviewer #1: Yes

Reviewer #2: Yes

Reviewer #1: The research objective of this paper is to compare the impact of different UNet architecture modifications on geomorphological feature extraction, in particular their performance under different amounts of training data. The authors used three different datasets: agricultural terraces, mining steps, and valley fill surfaces, and tested a variety of UNet variants, including activation function substitution, residual concatenation, dilated convolution, SE modules, attention gates, and MobileNetV2 encoders. The experimental design covers different training sample sizes (50 to full data) and evaluates various performance metrics. Among them, the following elements need further modification.

1. the literature review section may not be in-depth enough, especially regarding the application of UNet variants in other domains, have similar studies been conducted and have the authors adequately cited the relevant literature?

2. the experimental part, although testing a variety of variants, lacks discussion on model complexity and computational efficiency, such as training time and resource consumption of different variants, which is important for selection in practical applications.

3. the description of the datasets may not be detailed enough, e.g. does the number of samples in each dataset, the distribution of categories, feature differences, etc. affect the interpretation of the results?

4. in the discussion section, the authors mention that the model performance tends to stabilise after 500 samples, but the reason is not explored in depth, is it related to the model capacity or data characteristics?

5. in the conclusion, it is mentioned that MobileNetV2 performs poorly with a small amount of data, but it is not discussed whether it is because the pre-training weights are not used, and whether the effect of migration learning is considered?

Reviewer #2: Your manuscript has made the main points of your study clear and the manuscript was well prepared. But my concerns are that

1. if you look at the table 4 to 6, you may find there are a big difference between the OA and the F1-Score. Did you use the independent datasets that were different from the one for building the models and there was no strong spatial connection between some samples in model built datasets and some other samples in validation datasets? How do you explain this difference that OA is extremely high but F1 is extremely low?

2. You still look at these tables from 4 to 6. You may also find the increment of OA was small for all configurations as the number of chips increased while the increment of F1 was a little bit high. What happened? Was it inappropriate to use OA to evaluate the accuracy of classified results? Why did you think the sample size of 500 is appropriate in the applications of Unet for classification?

By the way, check the line 512, the word ‘then’ was misspelled.

**Do you want your identity to be public for this peer review?** For information about this choice, including consent withdrawal, please see our Privacy Policy

Reviewer #1: No

Reviewer #2: **Yes: ** Jinlong Fan

---

## [Author Response · Author response to Decision Letter 1]

5 May 2025

Please see the response to reviewers file.

---

## [Decision Letter · Decision Letter 1]

Comparing UNet configurations for anthropogenic geomorphic feature extraction from land surface parameters

PONE-D-25-06225R1

Dear Dr. Maxwell,

We’re pleased to inform you that your manuscript has been judged scientifically suitable for publication and will be formally accepted for publication once it meets all outstanding technical requirements.

Kind regards,

Xiaoyong Sun

Academic Editor

PLOS ONE

sunx1@sdau.edu.cn

Additional Editor Comments (optional):

Reviewers' comments:

Reviewer's Responses to Questions

**Comments to the Author**

Reviewer #1: All comments have been addressed

2. Is the manuscript technically sound, and do the data support the conclusions?

Reviewer #1: Yes

3. Has the statistical analysis been performed appropriately and rigorously?

Reviewer #1: Yes

4. Have the authors made all data underlying the findings in their manuscript fully available?

Reviewer #1: No

5. Is the manuscript presented in an intelligible fashion and written in standard English?

Reviewer #1: (No Response)

Reviewer #1: (No Response)

**Do you want your identity to be public for this peer review?** For information about this choice, including consent withdrawal, please see our Privacy Policy

Reviewer #1: No

---

## [Editor Report · Acceptance letter]

PONE-D-25-06225R1

PLOS ONE

Dear Dr. Maxwell,

I'm pleased to inform you that your manuscript has been deemed suitable for publication in PLOS ONE. Congratulations! Your manuscript is now being handed over to our production team.

Kind regards,

on behalf of

Dr. Xiaoyong Sun

Academic Editor

PLOS ONE